# Constrained Multi-Objective Bayesian Optimization

**Diantong Li**
School of Data Science
Chinese University of Hong Kong, Shenzhen
Shenzhen, China
diantongli@link.cuhk.edu.cn

**Fengxue Zhang**
Department of Computer Science
University of Chicago
Chicago, IL, USA
zhangfx@uchicago.edu

**Chong Liu**
Department of Computer Science
University at Albany, State University of New York
Albany, NY, USA
cliu24@albany.edu

**Yuxin Chen**
Department of Computer Science
University of Chicago
Chicago, IL, USA
chenyuxin@uchicago.edu

## Abstract

Multi-objective Bayesian optimization has been widely adopted in scientific experiment design, including drug discovery and hyperparameter optimization. In practice, regulatory or safety concerns often impose additional thresholds on certain attributes of the experimental outcomes. Previous work has primarily focused on constrained single-objective optimization tasks or active search under constraints. We propose CMOBO, a sample-efficient constrained multi-objective Bayesian optimization algorithm that balances learning of the feasible region (defined on multiple unknowns) with multi-objective optimization within the feasible region in a principled manner. We provide both theoretical justification and empirical evidence, demonstrating the efficacy of our approach on various synthetic benchmarks and real-world applications.

## 1 Introduction

Multi-objective Bayesian optimization (MBO) is essential in scientific experiment design, such as drug discovery [Fromer and Coley, 2023] and hyper-parameter optimization [Gardner et al., 2019], where efficient exploration of experimental space is crucial. However, real-world applications often require meeting additional safety or regulatory thresholds. For instance, drug discovery must balance therapeutic effectiveness with safety standards [Mellinghoff and Cloughesy, 2022], and hyper-parameter optimization must avoid overfitting or violating constraints [Karl et al., 2023].

Previous studies have focused on single-objective constrained Bayesian optimization, unconstrained multi-objective Bayesian optimization, or constrained active search, leaving a gap in constrained multi-objective Bayesian optimization (CMOBO). Research in Constrained Bayesian Optimization (CBO) has primarily extended unconstrained problems with early work by Schonlau et al. [1998] and subsequent studies incorporating posterior sampling methods [Eriksson and Poloczek, 2021] and information-based approaches [Hernández-Lobato et al., 2014, Wang and Jegelka, 2017] to scale CBO and improve feasibility analysis [Hernández-Lobato et al., 2015, Perrone et al., 2019, Takeno et al., 2022]. The augmented Lagrangian framework further transformed constrained tasks into unconstrained ones [Gramacy et al., 2016, Picheny et al., 2016, Ariafar et al., 2019], though it often lacked guarantees on feasibility and regret. Recent advancements [Zhou and Ji, 2022, Lu and Paulson, 2022, Xu et al., 2023, Guo et al., 2023] focus on relaxed CBO objectives to ensure theoretical convergence. In Constrained Active Search, active learning for the level sets offers sample

Workshop on Bayesian Decision-making and Uncertainty, 38th Conference on Neural Information Processing Systems (NeurIPS 2024).

efficiency guarantees [Gotovos et al., 2013] but struggles with multiple unknown functions. Recent approaches [Malkomes et al., 2021, Komiyama et al., 2022] emphasize diversity but fall short of balancing learning constraints with objective optimization. Multi-objective Bayesian Optimization Golovin and Zhang [2020], Daulton et al. [2022], Suzuki et al. [2020] typically relies on the scalarization of objectives combined and resort to generalized expected improvements. A principled integrated treatment of the constraints and multiple objectives in Bayesian optimization remains challenging.

We propose a sample-efficient constrained multi-objective Bayesian optimization (CMOBO) algorithm that balances learning of level sets on multiple unknowns with multi-objective optimization within feasible regions. The insight is that we constrain the search space to areas with the potential of being feasible, while the random scalarization [Deng and Zhang, 2019, Golovin and Zhang, 2020] allows an efficient and theoretically justified acquisition within the region. We offer theoretical justification, together with empirical evidence on both synthetic benchmarks and real-world applications, demonstrating the effectiveness and efficiency of the proposed method.

## 2 Preliminaries and Problem Statement

### 2.1 Problem Statement

Let $[n]$ denote the set $\{1, 2, ..., n\}$ and let $[x]^+$ denote function $\max(0, x)$. For a vector $\mathbf{x}$, its $\ell_2$ norm is denoted by $\|x\|$. For any two vectors $\mathbf{x}_1, \mathbf{x}_2$, we use $\mathbf{x}_1 \leq \mathbf{x}_2$ to denote their element-wise comparisons. To improve the readability of this paper, we utilize the big O notation to omit constant terms in theoretical results.

Consider a constrained multi-objective optimization problem:

$$\max_{\mathbf{x} \in \mathfrak{X}} \quad F(\mathbf{x}) = [f_1(\mathbf{x}), ..., f_m(\mathbf{x})],$$
$$\text{s.t.} \quad G(\mathbf{x}) = [g_1(\mathbf{x}), ..., g_c(\mathbf{x})] \geq 0.$$

where $f_i$ and $g_j : \mathfrak{X} \to \mathbb{R}, \forall i \in [m], \forall j \in [c]$ are black-box functions, $\mathfrak{X} \subset \mathbb{R}^d$ is the search space of $F : \mathbb{R}^d \to \mathbb{R}^m$, $m$ is the number of objectives and $c$ is the number of constraints. The goal is to find the Pareto frontier $P$ of $F$. For the main paper, we assume the search space $\mathfrak{X}$ to be finite, and extend our discussion of the continuous and compact search space in Appendix D.

**Definition 2.1** (Pareto Front). Define the feasible region of this problem as $\mathfrak{F} = \{\mathbf{x} | \mathbf{x} \in \mathfrak{X} \text{ and } g_i(\mathbf{x}) \geq 0, \forall i \in [c]\}$. For points $\mathbf{x}_1, \mathbf{x}_2$, in the context of this problem, $\mathbf{x}_1, \mathbf{x}_2 \in \mathfrak{F}$. $\mathbf{x}_1$ is said to *dominate* $\mathbf{x}_2$ if **1)** $f_i(\mathbf{x}_1) \geq f_i(\mathbf{x}_2), \forall i \in [m]$ and **2)** $\exists j \in [m]$ s.t. $f_j(\mathbf{x}_1) > f_j(\mathbf{x}_2)$. Denote $\mathfrak{X}^* = \{\mathbf{x} \in \mathfrak{F} | \mathbf{x} \text{ is not } dominated \text{ by any point } \in \mathfrak{F}\}$, then the Pareto front is defined as $P = \{F(\mathbf{x}) | \mathbf{x} \in \mathfrak{X}^*\}$.

**Assumption 1** (Gaussian Process). *Following Srinivas et al. [2009] (shown in Definition B.5), we assume that objectives and constraint functions are drawn from their GP priors.*

### 2.2 Evaluation Metrics

**Definition 2.2** (Simple Hypervolume Regret). Let simple hypervolume (HV) regret at step $t$, $r_t$ be the difference between the real *hypervolume indicator* of the Pareto front and the current approximation of the Pareto front. Define $Y_t$ to be a set of observed objectives, s.t. $|Y_t| = t$

$$r_t = \mathcal{HV}_z(P) - \mathcal{HV}_z(Y_t \cap F(\mathfrak{F}))$$

where $F(\mathfrak{F})$ is the range of the objective in feasible area, and $\mathcal{HV}_z(Y)$ is defined as $\text{vol}(\{y | y \geq z, y \text{ is dominated by some point } \in Y\})$. Maximizing $\mathcal{HV}_z(Y_t \cap F(\mathfrak{F}))$ reflects the exploration of the Pareto Front since it cannot be greater than $\mathcal{HV}_z(P)$. We accordingly define the cumulative hypervolume regret as $\mathcal{R}_t = \sum_{t=1}^T r_t$.

To address the problem of constrained optimization, we introduce the metric simple constraint violation that evaluates the constraint violation of the algorithm.

**Definition 2.3** (Simple Constraint Violation). We define the simple constraint violation of constraint $g_j$ at the $t^{th}$ observation $\mathbf{x}_t$ as $v_{j,t} = [-g_j(\mathbf{x}_t)]^+$ where $[\cdot]^+ = \max(0, \cdot)$. We also define the cumulative constraint violation of the $j - th$ constraint as $\mathcal{V}_{j,T} = \sum_t^T v_{j,t}$.

To assess the ability of the algorithm of simultaneously exploring the Pareto front and keeping a low constraint violation, based on the above definitions, we define a metric that considers both constraint violation and Hypervolume Regret as constraint regret at step $t$ as $\mathcal{C}_t$, the minimum of the sum of simple HV regret and simple constraint violation of all objectives among steps before $t$, which was originally proposed in Xu et al. [2023]

**Definition 2.4** (Constraint Regret).

$$\mathcal{C}_t = \min_{\tau \in [t]} \left\{ r_\tau + \sum_{j=1}^{c} v_{j,\tau} \right\}$$

In the experiments, we normalize $r_\tau$ and $\sum_{j=1}^{c} v_{j,\tau}$ so that they are comparable.

## 3 Constrained Multi-Objective Bayesian Optimization

---
**Algorithm 1** Constrained Multi-Objective Bayesian Optimization (CMOBO)

---
1: **for** $t \in \{1, ..., T\}$ **do**
2:     **if** $\max_{\mathbf{x} \in \mathfrak{X}} \{\min_{j \in [c]} w_{j,t}(\mathbf{x})\} < 0$ **then**
3:         **Declare infeasibility.**
4:     **end if**
5:     For scalarization, sample $\theta_t$ uniformly from $\mathcal{S}_{m-1}^+$
6:     Optimize acquisition function:
        $\mathbf{x}_t \in \arg\max_{\mathbf{x} \in \mathfrak{X}} s_{\theta_t}(U_t(\mathbf{x}))$
        s.t. $w_{j,t}(\mathbf{x}) \geq 0, \forall j \in [c]$.
7:     Evaluate $F$ at $\mathbf{x}_t$.
8:     Update GP posterior with the incoming observations.
9: **end for**

---

In each step of the algorithm, we apply a random scalarization, $s_{\theta_t}$, to the upper confidence bound surrogates of $m$ objectives. We substitute the $c$ constraint functions $g_j$ with their upper confidence bound surrogates, following the approach of Xu et al. [2023], and maximize the scalarized function subject to the new constraints. In line 2, we solve an auxiliary optimization problem, the solution of which helps determine whether to declare infeasibility. It can also be used in the subsequent optimization of the acquisition. See A for further discussion.

**Scalarization.** In our multi-objective setting, we address the challenge of trading off multiple acquisition functions by applying a scalarization mapping $s_{\theta_t}$ as defined in (1), parameterized by a randomly drawn variable $\theta_t$ at each iteration. This *hypervolume scalarization*, introduced by Deng and Zhang [2019], Golovin and Zhang [2020], allows for the Monte Carlo estimator of the hypervolume and its estimation error. This approach enables a principled combination of optimizing objectives and considering unknown constraints. We extend this scalarization to the constrained optimization scenario with both theoretical guarantees and comprehensive empirical evidence of its efficiency. Here, we define the acquisition function in Algorithm 1 with scalarization of UCBs.

**Definition 3.1** (Scalarization function, Deng and Zhang [2019], Golovin and Zhang [2020]). The *hypervolume scalarization* is defined as

$$s_\theta(y) = \min_{i \in [m]} ([y_i/\theta_i]^+)^m \text{ s.t. } y, \theta \in \mathbb{R}^m. \tag{1}$$

Furthermore, it holds that

$$\mathcal{HV}_z(Y_t) = c_m \mathbb{E}_{\theta \sim \mathcal{S}_{k-1}^+} \left[ \max_{y \in Y_t \cap F(\mathfrak{F})} s_\theta(y - z) \right] \tag{2}$$

where $\theta \sim \mathcal{S}_{k-1}^+$ denote drawing $\theta$ uniformly from $\mathcal{S}_{k-1}^+ = \{y \in \mathbb{R}^m | \|y\| = 1, y \geq 0\}$ and $c_m = \frac{\pi^{\frac{m}{2}}}{2^m \Gamma(\frac{m}{2}+1)}$.

This scalarization function is derived from the integration in the calculation of hypervolume, offering an unbiased estimation of the HV. When applying a sample of the random scalarization to the UCBs of the objectives at a certain time $t$, we have the following acquisition function.

**Definition 3.2** (Acquisition function). Let the $m$-dimensional vector $U_t(\mathbf{x})$ denote the UCB of $m$ objectives $f_i$. We define the acquisition function $\alpha_t(\mathbf{x})$ as the value of $U_t(\mathbf{x})$ scalarized by *hypervolume scalarization* (1).

$$U_t(\mathbf{x}) = (u_{f_1,t}(\mathbf{x}) - z_1, ..., u_{f_m,t}(\mathbf{x}) - z_m) \tag{3}$$

$$\alpha_t(\mathbf{x}) = s_{\theta_t}(U_t(\mathbf{x})) \tag{4}$$

where $z = (z_1, ..., z_m)$ is a chosen sub-optimal value.

Now, we need to construct an optimistic estimation of the constraint functions to incorporate the consideration of feasibility.

**Constrained optimization.** With the optimistic estimation of feasibility discussed above and the sample from the random scalarization adaptive tradeoff among multiple objectives, we can define the CMOBO optimization loop. In each iteration, we maximize the scalarized function subject to the newly defined constraints. In line 2 of Algorithm 1, we solve an auxiliary optimization problem to determine whether infeasibility should be declared. The solution to this auxiliary problem, $\arg\max_{\mathbf{x} \in \mathfrak{X}} \min_{j \in [c]} w_j(\mathbf{x})$, can also be leveraged in the optimization of the acquisition function, as it helps discard inactive constraints. Combined with the optimistic feasibility estimation in line 6 of Algorithm 1, we know that CMOBO iteratively picks the maximizer of the *UCB of the Monte Carlo estimator of the constrained hypervolume*. This allows the following theoretical guarantee of CMOBO.

## 4   Experiments

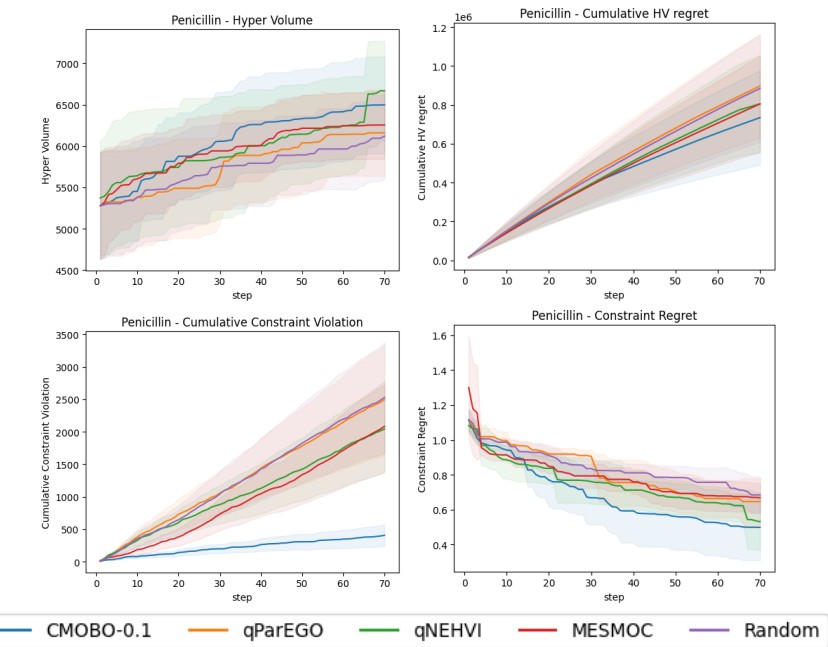

Figure 1: CMOBO performance on Penicillin function. From left to right: Hypervolume, Cumulative Hypervolume Regret, Cumulative Constraint Violation, Constraint Regret.

We applied Algorithm 1 to the following five tasks, including **Toy Function** ($d = 2, m = 2, c = 2$), **Branin-Currin Function** ($d = 2, m = 2, c = 2$), **Penicillin Function** ($d = 7, m = 3, c = 3$) Liang and Lai [2021], **Caco2++** ($d = 2175, m = 3, c = 3$) adapted from Park et al. [2024] and **ESOL+** ($d = 2133, m = 4, c = 4$) adapted from Delaney [2004]. Here, the Penicillin Function simulates penicillin production; the last two contain organic molecules and drug-related properties. Detailed experimental settings are in Appendix A. We trimmed the initial observations, leading to unequal starting values. Benchmarks include Parallel Noisy Expected Hypervolume Improvement Daulton et al. [2021] (**qNEHVI**), parallel ParEGO Daulton et al. [2020] (**qParEGO**), Max-value Entropy Search for Multi-Objective Bayesian Optimization with Constraints Belakaria et al. [2020] (**MESMOC**), and Random Search. All experiments except for **MESMOC** were carried out using the BoTorch Python library Balandat et al. [2020]. The complete results are in Appendix E.

**Trade-off Between Simple HV Regret and Cumulative Violation.** As shown in Figure 2 in Appendix E, while qNEHVI converges on simple HV regret at a rate comparable to CMOBO and achieves a higher final value, it incurs significantly more constraint violations. In more complex objectives, qNEHVI's simple HV performance is near-random, though its constraint violations remain sub-random. Scalarization-based **qParEGO** surpasses random search in Simple HV regret at around 30 steps, with near-random overall constraint violation performance. In this case, the Constraint Violation performance is dominated by Simple HV regret. This trade-off, measured by constraint regret (Definition 2.4), reflects the ability to explore the Pareto Front within feasible regions. High constraint regret indicates a failure to explore or maintain constraints. CMOBO outperforms all benchmarks, achieving the best balance with faster constraint regret reduction.

## 5 Theoretical Results

In this section, we provide the theoretical analysis of our algorithm CMOBO. We leverage the maximum mutual information gain $\gamma_{i,T}$ of the $i^{th} objective$ on GP after $T$ iterations, and $\gamma_T$ is an upper bound for $\gamma_{i,T}, \forall i \in [m]$ and $\gamma_{j,T}, \forall j \in [c]$. Detailed definitions are shown in B.6. The corresponding upper bounds for common kernels are previously studied by Srinivas et al. [2009].

**Case 1: $\mathfrak{F} \neq \emptyset$.** With assumptions stated in Section 2.1, given $\mathfrak{F} \neq \emptyset$ and the existence of the Pareto Front, we could bound the cumulative regret and corresponding violation in the following, with detailed proof deferred to Appendix C.

**Theorem 1** (Cumulative HV regret bound). *After $T$ iterations, under the conditions in Lemma 1, for $\delta \in (0,1)$ our Algorithm CMOBO satisfies that*

$$\mathcal{R}_T \leq O(m^2[\gamma_T T \ln T]^{1/2}) \tag{5}$$

*with probability at least $1 - \delta$.*

**Theorem 2** (Cumulative constraint violation bound). *Let $c$ denote the number of constraints, then $\forall j \in [c]$ and under the conditions in Lemma 1, with probability at least $1 - \delta$, our Algorithm CMOBO satisfies that*

$$\mathcal{R}_T \leq O(\sqrt{T \ln T \gamma_{j,T}}). \tag{6}$$

Combining Theorem 2 and Theorem 1, we can bound Constraint Regret defined in Definition 2.4 by $O(cm^2[\gamma_T \ln T/T]^{1/2})$ by taking the sum of two cumulative terms and taking minimum with respect to step indexes.

**Case 2: $\mathfrak{F} = \emptyset$.** Now we assume $\mathfrak{F} = \emptyset$. We conclude that we can declare infeasibility in (1) in Algorithm 1 within a certain number of steps with high probability.

**Theorem 3** (Declaration of infeasibility when the problem is infeasible). *With conditions in Lemma 1, and that $\lim_{T \to \infty} \frac{\sqrt{\ln T \gamma_T}}{\sqrt{T}} = 0$. If the problem is infeasible, i.e. $\exists j \in [c], \max_{\mathbf{x}} g_j(\mathbf{x}) = \epsilon < 0$ Then, given $\delta \in (0,1)$, Algorithm 1 will declare infeasibility within number of steps equivalent to $\bar{T} = \min_{T \in \mathbb{N}^+} \{T | \frac{\sqrt{\ln T \gamma_T}}{\sqrt{T}} \leq C\epsilon\}$ with probability at least $1 - \delta$.*

## 6 Conclusions

We proposed **CMOBO**, a stochastic scalarization-based Bayesian optimization algorithm based on stochastic scalarization for multiobjective constrained problems. **CMOBO** outperforms the **qParEGO** in various tests and excels in metrics that account for hypervolume regret and constraint violations, even if it does not always match **qNEHVI** in hypervolume regret convergence speed. We also provide a theoretical analysis of the metrics used. Future improvements include active learning of the feasible domain and adaptive scalarization parameters. While we have developed a heuristic to enhance candidate selection efficiency, challenges such as model misspecification remain.

## Acknowledgment

This work was partially done when CL was at the University of Chicago. This work was supported in part by the National Science Foundation under Grant No. IIS 2313131, IIS 2332475 and CMMI 2037026, and UAlbany Computer Science Department startup funding. The authors acknowledge the University of Chicago's Research Computing Center for their support of this work.

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

## A  Experimental Settings

**Baselines.**  We follow the tutorial for **qNEHVI** and **qParEGO** in https://botorch.org/tutorials/constrained_multi_objective_bo to implement the benchmarks.

**qNEHVI** Daulton et al. [2021] computes the expectation of $\mathcal{HV}_z(Y_t)$ w.r.t. $F(\mathbf{x}_t)$, expressed as $E_{F(\mathbf{x}_t)}[\mathcal{HV}_z(Y_t)]$. And deals with constraint by taking an conditional expecation: $\mathbf{x}_t = \arg\max E_{F(\mathbf{x})}[\mathcal{HV}_z(Y_t)|\mathbf{x} \in \mathfrak{F}]$.

**qParEGO** Daulton et al. [2020] applys an random augmented Chebychev scalarization $s_{\theta_t}(y) = \min_{i \in [m]} \theta_{i,t}(y_i - z_i)$ in each step to the objectives and uses GP to model the scalarized outcomes Knowles [2006] denoted as $\bar{s}_{\theta_t}(\mathbf{x})$. Then it applies conditional expected improvement (**EI**) to surrogate of scalarized objective.

$$\mathbf{x}_t = \arg\max E_{F(\mathbf{x})}\left[[\bar{s}_{\theta_t}(\mathbf{x}) - s_{\theta_t}^*]^+|\mathbf{x} \in \mathfrak{F}\right] = \arg\max E_{F(\mathbf{x})_{\theta_t}}\left[[\bar{s}_{\theta_t}(\mathbf{x}) - s_{\theta_t}^*]^+ \times \mathbb{I}(\mathbf{x} \in \mathfrak{F})\right]$$

where $\mathbb{I}(\mathbf{x})$ is an indicator for $\mathbf{x}$ being feasible, also approximated by surrogates of $g_j$, $s_{\theta_t}^*$ is the best observation of scalarized objective in current step.

**Inner Loop Optimization when Generalized to Continuous Search Space.** Some of the test objectives has continuous search space, so we can leverage the information of the maxima of auxiliary function proposed in Line 2 of Algorithm 1. BoTorch uses L-BFGS algorithm to conduct the optimization with non-linear constraints in Line 6 if the search space is continuous, which requires sufficient initial candidates of feasible solutions. The default candidate sampler of BoTorch cannot always generate sufficient candidates due to the complexity of constraints. We show in the analysis(Theorem 4) that the solutions of the auxiliary problem can be used as the initial candidates for L-BFGS with high probability. We will provide a discussion in D to show the algorithm could be applied to general continuous objectives.

**To Match The Parallel Settings**. Though the benchmark algorithms were designed for batched-output, we take number of queries $q = 1$ in each step to make them comparable to our approach, which is also seen in the experiments in Daulton et al. [2021]. We can still benefit from the parallel evaluation in optimizing the acquisition function.

**Penicillin Function.** The objective was proposed in Liang and Lai [2021]. We add a Gaussian noise with standard deviation 0.05 to the observations. We define the constraint to make penicillin production $\geq 10$, $CO_2$ production $\leq 60$ and reaction time $\leq 350$.

We take $\beta_{j,t} = c \log(t+1)$, where $c = 0.1$ or $c = 0.05$. We used the RBF kernel

$$k_{\text{RBF}}(\mathbf{x}_1, \mathbf{x}_2) = a \cdot \exp\left\{\frac{\|\mathbf{x}_1 - \mathbf{x}_2\|^2}{b}\right\}$$

We fit the GP models' parameters in each step in all experiments.

Since the distribution of objective is highly skewed, we observed high over-fitting of GP model even with large training data. To fix this, we apply a filtering trick to the initial samples of the GP model of each benchmark. We apply Voxel Grid Sampling trick to 64 randomly sampled candidates to get around 20 samples uniformly distributed in the range of initial candidate set in each trial. That would make the performances of all benchmarks differentiable to random search.

**Toy Function.** The objective is defined as:

$$F(\mathbf{x}_1, \mathbf{x}_2) = \left(-\frac{1}{\mathbf{x}_1} - \mathbf{x}_2, -\mathbf{x}_1 - \mathbf{x}_2{}^2\right)$$

$$\text{s.t. } \mathbf{x}_1, \mathbf{x}_2 \in [1, 1.5],$$

$$-\frac{1}{\mathbf{x}_1} - \mathbf{x}_2 \geq -1.9,$$

$$-\mathbf{x}_1 - \mathbf{x}_2{}^2 \geq -2.25$$

$\beta_{j,t} = 0.4 \log(4 \cdot (1+t))$. We used Matérn kernel for the GP model with 0.05 standard deviation noise. We took 10 random initial candidates in each trial.

$$k_{\text{Matérn}}(\mathbf{x}_1, \mathbf{x}_2) = \frac{2^{1-\nu}}{\Gamma(\nu)}\left(\sqrt{2\nu}d\right)^\nu K_\nu\left(\sqrt{2\nu}d\right)$$

- $d = (\mathbf{x}_1 - \mathbf{x}_2)^\top \Theta^{-2}(\mathbf{x}_1 - \mathbf{x}_2)$ is the distance between $\mathbf{x}_1$ and $\mathbf{x}_2$, scaled by the lengthscale parameter $\Theta$.
- $\nu$ is a smoothness parameter that takes values $\frac{1}{2}$, $\frac{3}{2}$, or $\frac{5}{2}$. Smaller values correspond to less smoothness.
- $K_\nu$ is a modified Bessel function.

**Branin-Currin Function.** A 2-D objective consisting of a Branin function and a Currin function.

$$f_1 = 15\mathbf{x}_2 - \left(5.1 \cdot \frac{(15\mathbf{x}_1 - 5)^2}{4\pi^2} + \frac{5(15\mathbf{x}_1 - 5)}{\pi} - 5\right)^2 + \left(10 - \frac{10}{8\pi}\right)\cos(15\mathbf{x}_1 - 5)$$

$$f_2 = \left(1 - \exp\left(-\frac{1}{2\mathbf{x}_2}\right)\right) \cdot \left(\frac{2300\mathbf{x}_1^3 + 1900\mathbf{x}_1^2 + 2092\mathbf{x}_1 + 60}{100\mathbf{x}_1^3 + 500\mathbf{x}_1^2 + 4\mathbf{x}_1 + 20}\right)$$

$$\text{s.t. } \mathbf{x}_1, \mathbf{x}_2 \in [0, 1]$$
$$f_1 \geq -20,$$
$$f_2 \geq -6$$

$\beta_{j,t} = 0.4 \log(4 \cdot (1+t))$. We use Matérn kernel with 0.01 standard deviation and 10 random initial candidates in each trial.

**Caco-2++.** The original version **Caco-2+**(d = 2133, m = 3) was proposed in Park et al. [2024], whose objective contains *permeability*, an experimentally tested value and 2 extra objectives, CrippenClogP, TPSA. The search space contains 906 drug molecules. We modified this dataset to come up with **Caco-2++**(d = 2175, m = 3, c = 3). For the domain, we augmented a new feature, mqn feature, to the domain of **Caco-2+**. We changed the objectives to *permeability*, TPSA, drug-likeliness score(QED). The search space contains 909 molecules. We constrain the objectives so that QED $\geq 0.5$, TPSA $\geq 80$, *permeability* $\geq -5$.

$\beta_{j,t} = c \log(1+t)$, $c = 0.1$ or $0.05$. We used Tanimoto kernel specialized for molecule representation. We take a 0.01 standard deviation observation noise and 64 initial candidates in each trial.

$$k_{\text{Tanimoto}}(\mathbf{x}_1, \mathbf{x}_2) = a \cdot \frac{\mathbf{x}_1 \cdot \mathbf{x}_2}{\|\mathbf{x}_1\|^2 + \|\mathbf{x}_2\|^2 - \mathbf{x}_1 \cdot \mathbf{x}_2}$$

**ESOL+.** The original **ESOL** (Delaney [2004]) dataset contains 1,144 organic molecules and an experimentally measured metric called log*(Solubility)*. We added three additional objectives—LogP, TPSA, and QED—to the original objective. For the domain, we use the same molecule representation, *fragprint*, as was used by **Caco-2+** ($d = 2133$). We constrain the objectives so that LogP $\geq 2.5$, QED $\geq 0.5$, TPSA $\geq 55$, and log*(Solubility)* $\geq -4$.

$\beta_{j,t} = c \log(1+t)$, $c = 0.1$ or $0.05$. We used Tanimoto kernel and a 0.005 standard deviation observation noise and 64 initial candidates in each trial.

# B   Definitions

**Definition B.1** (Scalarized Regret).

$$r_\theta(X_t) = \max_{\mathbf{x} \in \mathfrak{F}} s_\theta(F(\mathbf{x}) - z) - \max_{\mathbf{x} \in X_t \cap \mathfrak{F}} s_\theta(F(\mathbf{x}) - z) \tag{7}$$

**Definition B.2** (Bayes Regret). is the expectation of scalarized regret over all possible $\theta \sim \mathcal{S}_{k-1}^+$ :

$$R(X_t) = E_\theta[r_\theta(X_t)] \tag{8}$$

**Definition B.3** (Instantaneous Regret). is defined as:

$$r(\mathbf{x}_t, \theta_t) = \max_{\mathbf{x} \in \mathfrak{F}} s_{\theta_t}(F(\mathbf{x}) - z) - s_{\theta_t}(F(\mathbf{x}_t) - z) \tag{9}$$

where $\theta_t$ and $\mathbf{x}_t$ are the parameter and corresponding observation in the $t^{th}$ iteration of our algorithm.

**Definition B.4** (Cumulative Regret). is the cumulative sum of instantaneous regret:

$$R_C(T) = \sum_{t=1}^{T} r(\mathbf{x}_t, \theta_t) \tag{10}$$

**Definition B.5** (Gaussian Process Regression). Assume at time $t$ we query a new instance $(\mathbf{x}^t, y^t)$, with each entry $y_i^t \in y^t := (y_1^t, ..., y_m^t)$, equals $f_i^t(\mathbf{x}) + \delta_i^t$, where $\delta_i^t \stackrel{i.i.d}{\sim} Gaussian(0, \sigma^2), \forall i \in [m]$ models the noise of observation with fixed variance $\sigma^2$. Assume $f_i \sim \mathcal{GP}(0, k_i(\cdot, \cdot))$, where $k_i(\mathbf{x}, \mathbf{x}')$ is a kernel function that models the covariance of $f_i(.)$ at any two observations $\mathbf{x}, \mathbf{x}'$. A predictive distribution of $f_i(\mathbf{x})$ given $t$ observations is the posterior distribution: $f_i(\mathbf{x})|(y_i^1, ..., y_i^t) \sim Gaussian(\mu_{i,t}(\mathbf{x}), \sigma_{i,t}^2(\mathbf{x}))$. Like Srinivas et al. [2009], we further assume $k(\cdot, \cdot) \leq 1$.

- $\mu_{i,t}(\mathbf{x}) = k_i(\mathbf{x}_{1:t}, \mathbf{x})^T (K_i + \lambda I)^{-1} y_i^{1:t}$

- $\sigma_{i,t}^2(\mathbf{x}) = k_i(\mathbf{x}, \mathbf{x}) - k_i(\mathbf{x}_{1...t}, \mathbf{x})^T (K_i + \lambda I)^{-1} k_i(\mathbf{x}_{1...t}, \mathbf{x})$

- $K_i = [k_i(\mathbf{x}, \mathbf{x}')_{\mathbf{x}, \mathbf{x}' \in \{\mathbf{x}^1, ..., \mathbf{x}^t\}}] \in \mathbb{R}^{t \times t}$

- $k_i(\mathbf{x}_{1:t}, \mathbf{x}) = [k_i(\mathbf{x}_1, \mathbf{x}), ..., k_i(\mathbf{x}_t, \mathbf{x})]^T$

- $\lambda = \sigma^2$

Similarly, we define the posterior mean and variance function for constraint functions $g_j, \forall j \in [c]$ as $\mu'_{j,t}(\mathbf{x})$ and $\sigma_{j,t}^2(\mathbf{x})$.

**Definition B.6** (Maximum Mutual Information). The information gain of of a set of newly sampled points $A_t$ whose corresponding set of objective values is $Y_t$, is the mutual information of the distribution of $f_i$ and the distribution of $Y_t$. Both $f_i$ and $Y_t$ follow the assumption of noise observation and Gaussian process with kernel $k(\cdot, \cdot)$ in B.5.

$$I(Y_t; f_i) = H(f_i) - H(f_i|Y_t) = H(Y_t) - H(Y_t|f_i) \tag{11}$$

where $H(\cdot)$ is the Shannon entropy function. The closed form of $I(Y_t; f_i)$ is given by Srinivas et al. [2009]: $\frac{1}{2} log \det(I + \lambda^{-1} K_{i,t})$ where $K_{i,t} = [k_i(\mathbf{x}, \mathbf{x}')_{\mathbf{x}, \mathbf{x}' \in A_t}]$. Accordingly, the maximum mutual information for objective $f_i$ given $t$ observations is defined as:

$$\gamma_{i,t} = \max_{A_t \subset \mathfrak{X} \ s.t. \ |A_t|=t} \frac{1}{2} log \det(I + \lambda^{-1} K_{i,t}) \tag{12}$$

Similarly, we define $\gamma_{j,t}$ for constraint functions $g_j, \forall j \in [c]$.

**Definition B.7** (Level-Space). for $i \in [m], j \in [c]$ define

$$l_{i,t}(\mathbf{x}) = \mu_{i,t-1}(\mathbf{x}) - \beta_{i,t}^{\frac{1}{2}}\sigma_{i,t-1}(\mathbf{x}) \tag{13}$$

$$p_{j,t}(\mathbf{x}) = \mu'_{j,t-1}(\mathbf{x}) - \beta_{j,t}^{\frac{1}{2}}\sigma_{j,t-1}(\mathbf{x}) \tag{14}$$

$$u_{i,t}(\mathbf{x}) = \mu_{i,t-1}(\mathbf{x}) + \beta_{i,t}^{\frac{1}{2}}\sigma_{i,t-1}(\mathbf{x}) \tag{15}$$

$$w_{j,t}(\mathbf{x}) = \mu'_{j,t-1}(\mathbf{x}) + \beta_{j,t}^{\frac{1}{2}}\sigma_{j,t-1}(\mathbf{x}) \tag{16}$$

where $\beta_{i,t}, \beta_{j,t}$ are defined in Lemma 1.

# C   Proofs for Finite Discrete Search Space

We assume the search space is finite as stated in Section 2.1.

## C.1   Theorems

*Proof of Theorem 1.* From **Lemma 5** in Golovin and Zhang [2020]:

$$r_t = \mathcal{HV}_z(P) - \mathcal{HV}_z(Y_t \cap F(\mathfrak{F})) = c_m R(X_t) \tag{17}$$

Furthermore, from basic arithmetic relationship:

$$R(X_t) \leq E_{\theta_t}[r(\mathbf{x}_t, \theta_t)] \tag{18}$$

**Lemma 6** in Golovin and Zhang [2020] shows $c_m L \leq m$ for $s_\theta$. Combining with Lemma 2, we obtain

$$\sum_{t=1}^{T} c_m R(X_t) = \sum_{t=1}^{T} \mathcal{HV}_z(P) - \mathcal{HV}_z(Y_T) \tag{19}$$

$$\leq c_m E[R_C(T)] \tag{20}$$

$$\leq O(m^2[\gamma_T T \ln(T)]^{1/2}) \tag{21}$$

with probability at least $1 - \delta$. $\qquad\square$

*Proof of Theorem 2.*

$$\mathcal{V}_{j,T} = \sum_{t=1}^{T} v_{j,t} \tag{22}$$

$$\leq \sum_{t=1}^{T} 2\beta_{i,t}^{1/2}\sigma_{i,t-1}(\mathbf{x}_t) \tag{23}$$

$$\leq 2\sqrt{TM\beta_{j,T}\gamma_{j,T}} \tag{24}$$

$$= O(\sqrt{T\ln T\gamma_{j,T}}) \tag{25}$$

The inequality in (23) directly comes from Lemma 3. The last inequality is the conclusion of Lemma 4. $\qquad\square$

*Proof of Theorem 3.* Given iteration $T$ and $\delta \in (0,1)$, suppose infeasibility has not yet been declared. It follows that

$$\min_{j \in [c]} w_{j,t}(\mathbf{x}'_t) \geq 0, \forall t = 1, ..., T \tag{26}$$

where $\mathbf{x}'_t = \arg\max_{\mathbf{x}} \min_{j \in [c]} w_{j,t}(\mathbf{x})$.

From the infeasibility of the problem, assume $j^*$ satisfies the condition for $j$ in (3), and assume

$$\mathbf{x}_t^* = \arg\max_{\mathbf{x} \in \mathfrak{X}} g_{j^*}(\mathbf{x}) \tag{27}$$

then

$$g_{j^*}(\mathbf{x}_t^*) = \epsilon \tag{28}$$

$$\geq \min_{j \in [c]} \max_{\mathbf{x} \in \mathfrak{X}} g_j(\mathbf{x}) \tag{29}$$

$$\geq \max_{\mathbf{x} \in \mathfrak{X}} \min_{j \in [c]} g_j(\mathbf{x}) \tag{30}$$

$$\geq \max_{\mathbf{x} \in \mathfrak{X}} \min_{j \in [c]} p_{j,t}(\mathbf{x}) \tag{31}$$

(30) comes from the fact that

$$\min_{x \in D_x} \max_{y \in D_y} g(x, y) \geq \max_{y \in D_y} \min_{x \in D_x} g(x, y)$$

for any function two-side bounded $g$. (31) holds with probability at least $1 - \delta$ (Lemma 1).

Combining (26) and (28) to (31) and the fact that $\epsilon < 0$, we have

$$\min_{j \in [c]} w_{j,t}(\mathbf{x}_t') \geq 0 > \epsilon \geq \max_{\mathbf{x} \in \mathfrak{X}} \min_{j \in [c]} p_{j,t}(\mathbf{x}) \geq \min_{j \in [c]} p_{j,t}(\mathbf{x}_t') \tag{32}$$

with probability at least $1 - \delta$. Furthermore,

$$0 < -\epsilon \leq \min_{j \in [c]} w_{j,t}(\mathbf{x}_t') - \min_{j \in [c]} p_{j,t}(\mathbf{x}_t') \tag{33}$$

By picking

$$J = \arg \min_{j \in [c]} p_{j,t}(\mathbf{x}_t') \tag{34}$$

we have

$$-\epsilon \leq w_{J,t}(\mathbf{x}_t') - p_{J,t}(\mathbf{x}_t') = 2\beta_{J,t}^{1/2} \sigma_{J,t-1}(\mathbf{x}_t') \tag{35}$$

What follows is similar to the proof of Theorem 5.1 in Xu et al. [2023], by taking the sum for each $t$, and Lemma 4 shows:

$$-\epsilon \leq \frac{2\sqrt{TM\beta_{J,T}\gamma_{J,T}}}{T} \leq O(\frac{\sqrt{\ln T \gamma_T}}{\sqrt{T}}) \tag{36}$$

Here, we leverage the fact $\beta_{J,T} \sim O(\ln T)$. Then, there exists a $\bar{C} > 0$ such that

$$-\epsilon \leq \bar{C} \frac{\sqrt{\ln T \gamma_T}}{\sqrt{T}} \tag{37}$$

By taking $C = -\frac{1}{\bar{C}}$,

$$C\epsilon \leq \frac{\sqrt{\ln T \gamma_T}}{\sqrt{T}} \tag{38}$$

with probability at least $1 - \delta$. For now, we have created a necessary condition for infeasibility not being declared until $T$ steps conditioning on problem itself being infeasible. By argument of contradiction: if infeasibily is never declared, there must exist a $C < 0$ such that (38) is true for all $T \in \mathbb{N}^+$. However, since $\lim_{T \to \infty} \frac{\sqrt{\ln T \gamma_T}}{\sqrt{T}} = 0$, such $C$ would never exist. $\square$

**Theorem 4** (Declaration of Infeasibility in the Feasible Case). *With assumptions in Lemma 1, if the problem is feasible, i.e.*

$$\max_{\mathbf{x} \in \mathfrak{X}} \min_{j \in [c]} g_j(\mathbf{x}) \geq 0 \tag{39}$$

*then, in each iteration,* $\mathbf{x}_t' = \arg \max_{\mathbf{x} \in \mathfrak{X}} \min_{j \in [c]} w_j(\mathbf{x})$ *in Algorithm 1 is feasible for set* $\{\mathbf{x} | w_{i,t}(\mathbf{x}) \geq 0, \forall i \in [c]\}$, *which is equivalent to infeasibility not being declared, with probability $\geq 1 - \delta$.*

*Proof of Theorem 4.* Until the $T^{th}$ iteration, given $\delta \in (0, 1)$ from Lemma 1, we conclude that the following holds with probability at least $1 - \delta$

$$w_{j,t}(\mathbf{x}) \geq g_j(\mathbf{x}) \tag{40}$$
$$\forall \mathbf{x} \in \mathfrak{X}, \forall t \in \{1, ..., T\}, \forall i \in [c]$$

then, it follows that

$$\max_{\mathbf{x} \in \mathfrak{X}} \min_{j \in [c]} w_{j,t}(\mathbf{x}) \geq \max_{\mathbf{x} \in \mathfrak{X}} \min_{j \in [c]} g_{j,t}(\mathbf{x}) \geq 0 \tag{41}$$

with probability at least $1 - \delta$. Then, let $\mathbf{x}_t' = \arg \max_{\mathbf{x} \in \mathfrak{X}} \min_{j \in [c]} w_{j,t}(\mathbf{x})$, we have $w_{j,t}(\mathbf{x}_t') \geq 0, \forall j \in [c]$. $\square$

## C.2 Lemmas

**Lemma 1** (Lemma 5.1, Srinivas et al. [2009] ). *Let cardinality of $\mathfrak{X}$, $|\mathfrak{X}| < \infty$. Then, $\forall \delta \in (0, 1)$*

$$|\mu_{i,t-1}(\mathbf{x}) - f_i(\mathbf{x})| \leq \beta_{i,t}^{1/2} \sigma_{i,t-1}(\mathbf{x}) \tag{42}$$

$$|\mu'_{j,t-1}(\mathbf{x}) - g_j(\mathbf{x})| \leq \beta_{j,t}^{1/2} \sigma_{j,t-1}(\mathbf{x}) \tag{43}$$

$$\forall \mathbf{x} \in \mathfrak{F}, \forall i \in [m], \forall j \in [c], \forall t \in \{1, ..., T\}$$

*with probability $\geq 1 - \delta$. $\mu_{i,t-1}, \mu'_{j,t}, \sigma_{i,t-1}, \sigma_{j,t-1}$ and $\sigma$, are defined in B.5. $\mu_{0,0}$ and $\sigma_{0,0}$ are prior mean and standard deviation. $\beta_{i,t}, \beta_{j,t}$ are defined as $2 \log\left((m + c)|\mathfrak{X}|\pi_t/\delta\right), \forall i \in [m], j \in [c]$. And $\pi_t = \frac{\pi^2 t^2}{6}$.*

**Lemma 2** (Modified version of Theorem 7 in Golovin and Zhang [2020]). *In our algorithm, suppose $s_\theta(y)$ is L-Lipschitz for all possible $\theta$. With conditions in Lemma 1, the expected cumulative regret (10) is bounded*

$$E[R_C(T)] = O(Lm[\gamma_T T ln(T)]^{1/2}) \tag{44}$$

*with probability at least $1 - \delta$ for any $\delta \in (0, 1)$. where $\gamma_T$ is an upper bound for $\gamma_{k,T}$.*

*Proof of Lemma 2.* The unconstrained and definite version of this lemma was directly used in Golovin and Zhang [2020] and was proved by Paria et al. [2019]. Define $\mathfrak{F}_t = \{\mathbf{x} \mid w_{j,t}(\mathbf{x}) \geq 0, \forall j \in [c]\}$. Let's pick $\mathbf{x}_t^* = \arg\max_{\mathbf{x} \in \mathfrak{F}_t} s_{\theta_t}(F(\mathbf{x}))$ and $\mathbf{x}_t = \arg\max_{\mathbf{x} \in \mathfrak{F}_t} s_{\theta_t}(U_t(\mathbf{x}))$ in our problem. Then it follows that

$$E[R_C(T)] = E\left[\sum_{t=1}^{T}\left(\max_{\mathbf{x} \in \mathfrak{F}} s_{\theta_t}(F(\mathbf{x})) - s_{\theta_t}(F(\mathbf{x}_t))\right)\right] \tag{45}$$

$$\leq \underbrace{E\left[\sum_{t=1}^{T} s_{\theta_t}(U_t(\mathbf{x}_t)) - s_{\theta_t}(F(\mathbf{x}_t))\right]}_{B_1} \tag{46}$$

$$+ \underbrace{E\left[\sum_{t=1}^{T} s_{\theta_t}(F(\mathbf{x}_t^*)) - s_{\theta_t}(U_t(\mathbf{x}_t^*))\right]}_{B_2} \tag{47}$$

(46) holds if $s_{\theta_t}(U_t(\mathbf{x}_t)) - s_{\theta_t}(U_t(\mathbf{x}_t^*)) > 0$ and $s_{\theta_t}(F(\mathbf{x}_t^*)) - \max_{\mathbf{x} \in \mathfrak{F}} s_{\theta_t}(F(\mathbf{x})) > 0$. The first condition is always true by the definition of $\mathbf{x}_t$. From Lemma 1, we know $w_{j,t}(\mathbf{x}) \geq g_j(\mathbf{x}), \forall j \in [c], t \in \{1, ..., T\}, \mathbf{x} \in \mathfrak{X}$ with probability at least $1 - \delta$, then $\mathfrak{F}_t \subset \mathfrak{F}$ with probability at least $1 - \delta$, which means the second condition holds with the same probability.

Given (46), for finite $\mathfrak{X}$: Lemma 3 in Paria et al. [2019] shows

$$B_1 \leq O(L\left[m^2 T \beta_T \gamma_T\right]^{1/2}) + O(Lm \sum_{t=1}^{T} \exp(-\frac{\beta_{i,t}}{2})) \tag{48}$$

where $\beta_T$ is an upper bound for $\beta_{i,T}, \forall i \in [m]$. Lemma 2 in Paria et al. [2019] shows

$$B_2 \leq O(Lm \sum_{t=1}^{T} \sum_{\mathbf{x} \in \mathfrak{X}} \exp(-\frac{\beta_{i,t}}{2})) \tag{49}$$

The final bound comes from the fact that $\beta_{i,t} \sim O(\ln t)$ and $\exp(\frac{-\beta_{i,t}}{2}) \sim O(\frac{1}{t^2})$ and that $\sum_{t=1}^{T} \frac{1}{t^2} = \frac{\pi^2}{6}$. $\qquad \square$

**Lemma 3** (Lemma 4.1 in Xu et al. [2023]). *With the conditions in Lemma 1. With probability $1 - \delta$,*

$$v_{j,t} \leq 2\beta_{j,t}^{1/2} \sigma_{j,t-1}(\mathbf{x}_t), \ \forall j \in [c], \forall t \in \{1, ..., T\} \tag{50}$$

**Lemma 4** (A corollary of Lemma 5.4 in Srinivas et al. [2009]). *$\forall j \in [c]$, with $\mathbf{x}_1, ..., \mathbf{x}_T$ selected by our algorithm,*

$$\sum_{t=1}^{T} 2\beta_{j,t}^{1/2} \sigma_{j,t-1}(\mathbf{x}_t) \leq 2\sqrt{TM\beta_{j,T}\gamma_{j,T}} \tag{51}$$

*where $M = \frac{1}{\log(1+\sigma^{-2})}$, $\sigma^2$ is the variance of the gaussian noise defined in Definition B.5.*

## D    Discussion of Continuous and Compact Search space

### D.1    Assumptions

We now consider $\mathfrak{X}$ to be continuous and compact. WLOG, $\mathfrak{X} := [0,1]^d$. We keep the assumptions for $\mathcal{GP}$s in Assumption 1.

### D.2    Generalized CMOBO

In order to provide theoretical justification for our algorithm CMOBO with continuous $\mathfrak{X}$, we provide a modified algorithm which is theoretical sound for both finite and infinite search space.

In each step $t$, we consider $\bar{\mathfrak{X}}_t$ as a finite discretization of $\mathfrak{X}$. $\bar{\mathfrak{X}}_t$ constains points evenly distributed in $\mathfrak{X}$ with $\tau_t^{-1}$ being the distance between any two adjacent points in $\bar{\mathfrak{X}}_t$. Denote $[\mathbf{x}]_t$ as the closest point in $\tilde{\mathfrak{X}}_t$ to $\mathbf{x}$.

We also re-define the upper and lower confidence bounds for $f_i$ and $g_j$ in Definition B.7.

**Definition D.1** (Modified confidence bound). for $i \in [m], j \in [c]$ define

$$l_{i,t}(\mathbf{x}) = \mu_{i,t-1}([\mathbf{x}]_t) - \beta_{i,t}^{\frac{1}{2}}\sigma_{i,t-1}([\mathbf{x}]_t) - \frac{1}{t^2} \tag{52}$$

$$p_{j,t}(\mathbf{x}) = \mu'_{j,t-1}([\mathbf{x}]_t) - \beta_{j,t}^{\frac{1}{2}}\sigma_{j,t-1}([\mathbf{x}]_t) - \frac{1}{t^2} \tag{53}$$

$$u_{i,t}(\mathbf{x}) = \mu_{i,t-1}([\mathbf{x}]_t) + \beta_{i,t}^{\frac{1}{2}}\sigma_{i,t-1}([\mathbf{x}]_t) + \frac{1}{t^2} \tag{54}$$

$$w_{j,t}(\mathbf{x}) = \mu'_{j,t-1}([\mathbf{x}]_t) + \beta_{j,t}^{\frac{1}{2}}\sigma_{j,t-1}([\mathbf{x}]_t) + \frac{1}{t^2} \tag{55}$$

$\mu_{i,t-1}, \mu'_{j,t}, \sigma_{i,t-1}, \sigma_{j,t-1}$ are defined in B.5. $\mu_{0,0}$ and $\sigma_{0,0}$ are prior mean and standard deviation. $\beta_{i,t} = 2\log\left(2\pi_t(m+c)/\delta\left[dtb_i\log\left(2da_i(m+c)/\delta\right)\right]^d\right)$, $\beta_{j,t} = 2\log\left(2\pi_t(m+c)/\delta\left[dtb_j\log\left(2da_j(m+c)/\delta\right)\right]^d\right)$ for some constants $a_i, a_j, b_i, b_j > 0$ and $\pi_t = \frac{\pi^2 t^2}{6}$. $\tau_t = dt^2 B\sqrt{\log(2dA(m+c)/\delta)}$. Detailed definition of $A, B$ are in Paria et al. [2019], B.2.

We use a new lemma with equal function to Lemma 1.

**Lemma 5** (Lemma 5.7 in Srinivas et al. [2009]). *Under the assumptions stated in Section D.1 (the following discussions are also based on these two assumptions), $\forall \delta \in (0,1)$,*

$$|\mu_{i,t-1}([\mathbf{x}]_t) - f_i(\mathbf{x})| \leq \beta_{i,t}^{\frac{1}{2}}\sigma_{i,t-1}([\mathbf{x}]_t) + \frac{1}{t^2} \tag{56}$$

$$|\mu'_{j,t-1}([\mathbf{x}]_t) - g_j(\mathbf{x})| \leq \beta_{i,t}^{\frac{1}{2}}\sigma_{j,t-1}([\mathbf{x}]_t) + \frac{1}{t^2} \tag{57}$$

$$\forall \mathbf{x} \in \mathfrak{F}, \forall i \in [m], \forall j \in [c], \forall t \in \{1, ..., T\}$$

*with probability $\geq 1 - \delta$. The parameters follow D.1.*

We will show a modified Lemma 2 still holds with Definition B.7 replaced by Definition D.1 and Lemma 1 replaced by Lemma 5.

**Lemma 6** (A modified version of Lemma 2). *In our algorithm, suppose $s_\theta(y)$ is L-Lipschitz for all possible $\theta$. With the conditions in Definition D.1, Lemma 5. For $\delta \in (0,1)$, the expected cumulative regret (10) is bounded with probability at least $1 - \delta$*

$$E[R_C(T)] = O(Lmd^{1/2}[\gamma_T T ln(T)]^{1/2}) \tag{58}$$

*where $\gamma_T$ is defined in Definition B.6.*

*Proof of Lemma 6.* We take a different approach to split the target into three parts:

$$E[R_C(T)] = E\left[\sum_{t=1}^{T}\left(\max_{\mathbf{x}\in\mathfrak{F}} s_{\theta_t}(F(\mathbf{x})) - s_{\theta_t}(F(\mathbf{x}_t))\right)\right] \tag{59}$$

$$\leq \underbrace{E\left[\sum_{t=1}^{T} s_{\theta_t}(U_t(\mathbf{x}_t)) - s_{\theta_t}(F(\mathbf{x}_t))\right]}_{B_1} \tag{60}$$

$$+ \underbrace{E\left[\sum_{t=1}^{T} s_{\theta_t}(F([\mathbf{x}_t^*]_t)) - s_{\theta_t}(U_t([\mathbf{x}_t^*]_t))\right]}_{B_2} \tag{61}$$

$$+ \underbrace{E\left[\sum_{t=1}^{T} s_{\theta_t}(F(\mathbf{x}_t^*)) - s_{\theta_t}(F([\mathbf{x}_t^*]_t))\right]}_{B_3} \tag{62}$$

if $s_{\theta_t}(U_t(\mathbf{x}_t)) \geq s_{\theta_t}(U_t(\mathbf{x}_t^*))$ and $s_{\theta_t}(U_t(\mathbf{x}_t)) \geq s_{\theta_t}(U_t([\mathbf{x}_t^*]_t))$ and $\max_{\mathbf{x}\in\mathfrak{F}_t} s_{\theta_t}(F(\mathbf{x})) \geq \max_{\mathbf{x}\in\mathfrak{F}} s_{\theta_t}(F(\mathbf{x}))$. The first two conditions are definitely true by definition of $\mathbf{x}_t$ and $\mathbf{x}_t^*$ in Theorem 2. The last one holds if $\mathfrak{F} \subset \mathfrak{F}_t$. With a similar argument, that still holds with probability at least $1 - \delta$. Take $\beta_T$ as an upper bound of $\beta_{i,T}, \forall i \in [m]$. By Lemma 3 in Paria et al. [2019],

$$B_1 \leq O(Lm\,(T\beta_T\gamma_T)^{1/2}) + O(Lm\sum_{t=1}^{T}\exp(-\beta_{i,t}/2)) \tag{63}$$

By Lemma 2 in Paria et al. [2019]

$$B_2 \leq O(Lm\sum_{t=1}^{T}\exp(-\beta_{i,t}/2)) \tag{64}$$

By result in Ghosal and Roy [2006], if kernel $k$ is stationary and $4^{th}$-differentiable, $f_i, g_j \sim \mathcal{GP}_i, \mathcal{GP}_j$ respectively. $\exists a_i, b_j > 0$ s.t. $\forall k \in \{1, ..., d\}, \forall Q > 0$

$$\mathbb{P}\left(\sup_{\mathbf{x}}\left|\frac{df_i}{d\mathbf{x}_k}\right| > Q\right) \leq a_i\exp\left((-Q/b_i)^2\right) \tag{65}$$

$$\mathbb{P}\left(\sup_{\mathbf{x}}\left|\frac{dg_j}{d\mathbf{x}_k}\right| > Q\right) \leq a_j\exp\left((-Q/b_j)^2\right)$$

Here we define $A, B$ in Lemma 5: $A = \max\left\{\sup_{i\in[m]} a_i, \sup_{j\in[c]} a_j\right\}, B = \max\left\{\sup_{i\in[m]} b_i, \sup_{j\in[c]} b_j\right\}$. From equation 16 of Paria et al. [2019]

$$B_3 \leq O(\sum_{t=1}^{T} Lm\frac{dAB\sqrt{\pi}}{2\tau_t}) \tag{66}$$

$\square$

With Lemma 6, we can follow the idea of proof of Theorem 1 to bound the cumulative HV regret with continuous and compact search space by $O(m^2 d^{1/2}[\gamma_T T \ln T]^{1/2})$. Note that the cumulative HV regret bound for continuous and compact search space additionally considers the size of instance vector $d$, which arises from a dynamic discretization density $\tau_t$ defined in Definition D.1.

## E   Supplementary Experimental Results

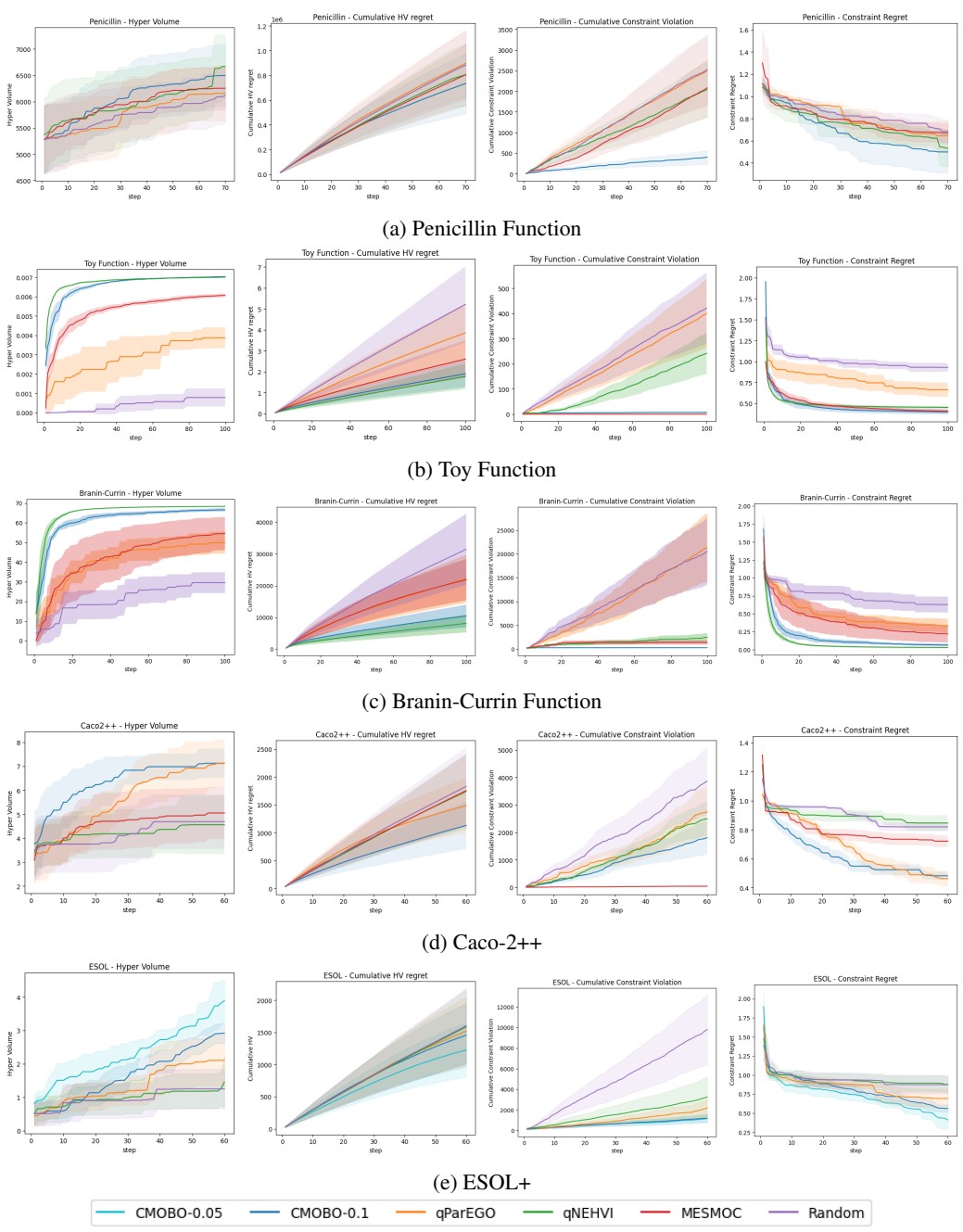

Figure 2: CMOBO performance on other objectives. From left to right: Hypervolume, Cumulative Hypervolume Regret, Cumulative Constraint Violation, Constraint Regret. Curves are shaded by area between ± 0.1 standard error.

