# OpenReview forum: "Constrained Multi-objective Bayesian Optimization"
_NeurIPS.cc/2024/Workshop/BDU — NeurIPS BDU Workshop 2024 Poster_

### Official Review · Reviewer_QE1b · 2024-09-26
**Reads well, but contributions should be toned-down and set-up clarified**

**Rating:** 6
**Confidence:** 3

**Review:**

# Summary

This paper proposes a method for *Constrained Multi-objective Bayesian Optimization* (CMOBO), based on previous work by Golovin & Zhang, 2020 and Deng & Zhang, 2019.

In a nutshell, the proposed method optimizes a random scalarization of the multivariate objective using the upper confidence bound. It is known that, by following this process, we end up optimizing the hypervolume improvement. The authors incorporate constraints by modelling them with GPs and choosing points such that the upper confidence bound of the constraints is positive.

The authors compare their CMOBO method against other well-known unconstrained alternatives, showing they can optimize multi-variate problems while violating the constraints less often.

# Review

The paper reads well and is well-motivated (indeed, practitioners in drug discovery and other applications need to optimize multiple objectives while satisfying certain constraints). However, the presentation of some of the content could be improved, some of the contributions should be toned down, and the experimental set-up and algorithm should be clarified further.

## On the contributions and other related work

Although not in our community, constrained multiobjective Bayesian Optimization was explored in the context of aerospace vehicle design:  https://hal.science/hal-02912982/document.

They take a similar approach to CEI (namely, multiplying by the probability that a constraint is satisfied). It might be interesting to benchmark against such a method in future versions of this paper.

## On the budget and experimental set-up

The choice of budget for most experiments feels unmotivated. Why stop at 70 for e.g. the Penicillin example?

Low budgets should be justifiable by talking to practitioners (who indeed sometimes rely on expensive experiments, for which they can afford ~100 evaluations). Still, it would be great to justify the choice of budget further.

## What are those 0.05 and 0.1?

Algorithm 1 has no inputs to it. We can see in the results, however, that there are two presentations of the algorithm: CMOBO-0.05 and CMOBO-0.1. What are the constants next to the name? Are they the UCB beta parameters? Could you clarify that further?

## On presentation

Minor details:

- The scale for cumulative HV regret could be presented in log-scale to make the differences clearer.
- The random scalarization is studied in Golovin and Zhang 2020, but it was originally proposed by Deng and Zhang in 2019. This might warrant citing Deng and Zhang as well.
- In the conclusions, I would change “we’ve” to “we have”.
- The name of the algorithm corresponds to a *meta name*. It is *not* the only algorithm for constrained multi-objective BO out there, so I would like to suggest a name change.

---

### Official Review · Reviewer_66n7 · 2024-09-26
**The paper introduces a Constrained Multi-Objective Bayesian Optimization (CMOBO) algorithm that efficiently balances learning the level sets of multiple unknowns with multi-objective optimization within feasible regions, using random scalarization. This approach addresses gaps in the literature by focusing on constrained optimization in a multi-objective context, where previous work primarily focused on single-objective constrained optimization or unconstrained multi-objective tasks. The proposed algorithm demonstrates its effectiveness through both theoretical analysis and empirical evidence across various benchmarks, including synthetic and real-world applications.  Key Contributions:  Novel Algorithm: CMOBO integrates constraint management with multi-objective optimization by leveraging random scalarization, ensuring a principled search within feasible regions. This innovation addresses the challenge of balancing constraint learning and optimization in high-dimensional, uncertain spaces. Empirical Validation: The algorithm shows superior performance, particularly in constraint handling and hypervolume regret reduction, compared to existing benchmarks like qNEHVI and qParEGO.  Theoretical Guarantees: The paper provides bounds on cumulative regret and constraint violation, enhancing the credibility of the approach.  Advantages:  Efficiency: CMOBO is highly sample-efficient, which is critical for real-world applications like drug discovery and hyperparameter optimization, where constraints like safety or regulatory limits are pivotal. Balanced Exploration and Optimization: The integration of learning feasible regions while optimizing multiple objectives fills a significant gap in constrained optimization literature.  Drawbacks:  Model Misspecification: The paper acknowledges challenges related to model misspecification, which could affect performance. This is particularly relevant when dealing with noisy, real-world datasets.  Scalarization Dependence: The reliance on random scalarization, while effective, may be limited in certain scenarios where adaptive approaches could improve results.  Overall, the paper makes a meaningful contribution to the field of Bayesian optimization, particularly by addressing the complexities of constrained multi-objective tasks. It stands out in comparison to existing methods by providing a well-rounded solution, though future work might focus on improving model robustness and exploring adaptive scalarization techniques.**

**Rating:** 8
**Confidence:** 4

**Review:**

-----

Evaluation:

1. Quality

The quality of this paper is credible, both in its technical execution and its depth of research. The authors present a well-defined and structured approach to Constrained Multi-Objective Bayesian Optimization (CMOBO). The mathematical rigor is evident by providing theoretical proofs that guarantee regret bounds for cumulative hypervolume and constraint violations.

Strengths in Quality:
Mathematical Rigor: The use of Gaussian Process (GP) priors, scalarization techniques, and the clear formulation of the Pareto front provide a solid foundation for the algorithm. The theoretical results are well-justified through proofs (e.g., the cumulative regret bound) and align with established literature in Bayesian optimization.
Empirical Validation: The algorithm is tested on synthetic functions and real-world problems (e.g., Penicillin production, molecular datasets), demonstrating its versatility and robustness.

Areas for Improvement in Quality:
Model Misspecification Handling: While the algorithm is robust in various scenarios, it relies heavily on the Gaussian Process assumption, which might be limiting in practical applications where GPs may not accurately model the data. Introducing methods to handle model misspecification would improve the reliability of the results.
Scalability to Higher Dimensions: While the method is tested on relatively high-dimensional problems, the scalability in extremely high-dimensional spaces, which are typical in real-world tasks, could be further explored and validated.


2. Clarity

The clarity of the paper is generally convincing, with clear explanations of the problem setup, the proposed algorithm, and the accompanying theoretical analysis. However, the clarity could be improved in some sections to better engage readers unfamiliar with the nuanced details of multi-objective optimization.

Strengths in Clarity:
Definitions and Problem Statement: The paper starts by clearly defining the constrained multi-objective problem with precise mathematical formulations. Definitions such as the Pareto frontier, hypervolume regret, and constraint violations are explained adequately.
Figures and Visualizations: The performance of CMOBO is well-illustrated through plots, comparing hypervolume regret and constraint violation against baselines (e.g., qNEHVI, qParEGO). These help in understanding the trade-offs involved in the optimization process.

Areas for Improvement in Clarity:
Technical Jargon: Some sections, particularly those involving detailed proofs or complex algorithms, could benefit from additional explanations or footnotes. Not all readers may have a deep background in GP-based optimization, and providing simpler overviews or intuitive explanations would increase accessibility.
Detailed Algorithm Explanation: While the algorithm is outlined, more focus on explaining its intuition and real-world applicability would benefit readers. A clearer breakdown of how the random scalarization interacts with the constraints in different settings could enhance comprehension.


3. Originality

The originality of the paper lies in its novel integration of random scalarization within the context of constrained multi-objective Bayesian optimization. This approach adds a unique angle to the existing literature, particularly in the way it tackles feasibility learning and optimization trade-offs simultaneously.

Strengths in Originality:
Unique Problem Addressing: Constrained multi-objective optimization is a relatively under-explored area compared to its unconstrained counterpart. The authors have contributed by addressing a significant gap in the literature: how to balance learning constraints with optimizing multiple objectives.
Scalarization Approach: The use of random scalarization to explore the feasible region is an innovative method. It allows the algorithm to adapt dynamically to the constraints while optimizing multiple objectives, which has not been widely explored in previous works.

Areas for Improvement in Originality:
Further Extension of Techniques: While the scalarization technique is a novel contribution, it is still somewhat based on established methods of GP optimization and scalarization in multi-objective contexts. Further innovation could come from adaptive or learning-based approaches to scalarization, where the parameters are adjusted based on real-time feedback from the problem space.


4. Significance

The significance of this work is adequate, particularly for practitioners and researchers working on constrained optimization in real-world applications such as drug discovery, material science, and hyperparameter optimization.

Strengths in Significance:
Practical Applications: The constrained optimization framework presented is particularly useful for fields where safety and regulatory constraints are critical. The case studies on penicillin production and molecular datasets highlight the practical applicability of the method.
Contribution to Multi-Objective Optimization Literature: This work addresses an important gap in constrained multi-objective optimization, offering a method that performs well in both theoretical and practical scenarios. The proposed algorithm could serve as a building block for future research and real-world optimization tasks.

Areas for Improvement in Significance:
Broader Applicability: While the paper tests the algorithm on a variety of datasets, demonstrating its application to larger-scale industrial problems or broader fields could significantly enhance its impact. Future work could focus on applying CMOBO to other complex domains.


List of Pros and Cons:

Pros:

- Novel Algorithm: Introduces a sample-efficient approach for constrained multi-objective Bayesian optimization, integrating random
  scalarization with constraint handling.
- Theoretical Rigor: The paper provides theoretical guarantees for cumulative hypervolume regret and constraint violations, making the
   algorithm both practical and mathematically sound.
- Real-World Applications: Demonstrates effectiveness on synthetic and real-world datasets, particularly in drug discovery and
   hyperparameter optimization.
- Balanced Exploration-Exploitation: The algorithm manages a good trade-off between exploring the feasible region and optimizing
  objectives, which is crucial for constrained problems.

Cons:

- Model Misspecification: The reliance on GP models may limit the applicability of the algorithm in scenarios where GPs do not accurately
  reflect the underlying data structure.
- Technical Complexity: The detailed mathematical derivations, while necessary, could make the paper less accessible to non-specialists or
   those without a strong background in Bayesian optimization.
- Scalability: Although the algorithm is shown to perform well on relatively high-dimensional problems, its performance on extremely high
  -dimensional spaces could be further validated.
- Potential Over-reliance on Scalarization: The random scalarization approach, while innovative, may have limitations in certain
  applications. More adaptive methods could offer better performance in dynamically changing problem spaces.


-------

---

### Decision · Program_Chairs · 2024-10-09

Accept (Poster)